# An Unusual Case of Torticollis: Split Cord Malformation with Vertebral Fusion Anomaly: A Case Report and a Review of the Literature

**DOI:** 10.3390/children9071085

**Published:** 2022-07-20

**Authors:** Dong Hyun Ye, Da Yeong Kim, Eun Jae Ko

**Affiliations:** Department of Rehabilitation Medicine, Asan Medical Center, University of Ulsan College of Medicine, 88, Olympic-ro 43-gil, Songpa-gu, Seoul 05505, Korea; ydh7342@gmail.com (D.H.Y.); dykim1102@gmail.com (D.Y.K.)

**Keywords:** torticollis, split cord malformation, vertebral fusion anomaly

## Abstract

We describe the exceptional case of spinal cord malformation, associating neurenteric cyst, and cervical vertebral malformation, initially presenting as torticollis. A 4-month-old child presented with torticollis to the right since birth. A cervical spine X-ray revealed suspicious findings of fusion anomaly, and a cervical spine CT showed extensive segmentation-fusion anomaly with an anterior and posterior bony defect in the C1–6 vertebrae. A cervical spine MRI revealed extensive segmentation-fusion anomaly with an anterior bony defect, and the spinal cord split forward and backward at the C3 level, showing two hemicords. The anterior half of the hemicord and dural sac extended to the right inferior side, towards the upper blind end of esophageal duplication, and the posterior half joined the hemicord at the back and C6 level. After multidisciplinary collaboration, follow-up and conservative treatment were planned. At 12 months, he had developmental delay, and torticollis showed little improvement. No neurological abnormalities have been observed. The patient plans to undergo surgery for the cervical spine fusion anomaly. Cervical spine X-rays should always be performed when assessing a patient with torticollis to rule out cervical vertebral segmentation anomalies, despite the rarity of the condition.

## 1. Introduction

Torticollis is a common disorder that can occur at any age but is particularly prevalent in pediatrics. The etiology of torticollis is diverse, spanning sternomastoid tumor, muscular torticollis, postural torticollis, atlantoaxial rotatory fixation, neoplasm, ocular torticollis, dystonia, and cervical vertebral segmentation anomaly [1]. Congenital failure to establish the normal structure of seven cervical vertebrae and nearby intervertebral discs results in cervical vertebral segmentation anomaly. Anomalies such as fusion of the cervical spine and vertebral misshaping can cause the symptom of torticollis [2,3]. Spinal dysraphism refers to a spectrum of disorders resulting from abnormal neural tube development during varying stages of embryogenesis. Split cord malformation (SCM) is a rare subtype of developmental spinal dysraphism, accounting for 3.8% of all spinal dysraphism, in which fusion of a single spinal cord is inhibited by a membranous, bony, or fibrous septum, resulting in two persisting hemicords bound by a single or duplicated dural sac [4,5,6]. Furthermore, SCM is often associated with pathologic anomalies such as spina bifida, vertebral dysplasias, low-lying conus, thick filum, lipoma, hydromyelia, myelomeningoceles, dermoid cyst, dermal sinus, scoliosis, and torticollis [7,8,9]. SCMs typically present in early childhood with manifestations of pain and neurologic deficits; however, some asymptomatic patients remain undiagnosed until adulthood [10]. Few cases of SCM with cervical segment involvement have been described to date. We present an exceptional case of SCM, associating neurenteric cyst, and cervical vertebral malformation, initially presenting as torticollis.

## 2. Case Report

### 2.1. Presenting Concerns and Past Medical History

A 4-month-old male was brought in for a pediatric rehabilitation medicine consultation for the symptom of torticollis since birth. Fetal ultrasound revealed congenital diaphragmatic hernia (contents: stomach, liver, intestine) and suspicion of a ventricular septal defect. He was delivered by cesarean section at 34 + 1 weeks, weighing 2370 g. He required intubation for approximately 3 min before being admitted to the neonatal intensive care unit for congenital diaphragmatic hernia and prematurity. Postnatal echocardiography confirmed proper heart function without significant intracardiac anomaly, with no abnormal findings on brain and spine ultrasounds. Postnatal surgical correction was performed for the left central diaphragmatic hernia with herniated stomach, small bowel loops, spleen, and pancreas into the posterior mediastinum and right hemithorax. However, open gastrojejunostomy was still required to treat insufficient dietary progress and extensive gastroesophageal reflux. Esophageal anastomosis with gastric replacement and primary duodenal and stomach repair were performed when the patient was 3 months old.

### 2.2. Physical Examination

Physical examination at 4 months revealed a head tilt to the right and left plagiocephaly. At rest, the head was tilted 10 degrees to the right, whereas during passive neck rotation, movement of 20 degrees to the right and 60 degrees to the left was possible. Passive tilting allowed for movement of 30 degrees to the right and 20 degrees to the left. There was no palpable neck mass, but cutaneous stigma (tufts of hair) was observed at the posterior neck at the C3 level. In addition, ptosis of the right eyelid and short philtrum were identified. Neurological examination revealed slightly hypotonic muscle tone, with no upper motor neuron sign observed. Only partial control of the neck was possible due to a developmental delay.

### 2.3. Diagnostic Assessment

Neck ultrasonography, performed as a work-up for the right torticollis, revealed symmetric bilateral sternocleidomastoid muscles with no abnormalities. Additional CT and MRI were performed after a cervical spine X-ray revealed suspicious findings of fusion anomaly (Figure 1). The cervical spine CT showed extensive segmentation-fusion anomaly with anterior and posterior bony defects in the C1–6 vertebrae (Figure 2). A cervical spine MRI revealed extensive segmentation-fusion anomaly with an anterior bony defect in the C-vertebrae (Figure 3A). From the C3 level, the spinal cord split forward and backward, showing two hemicords (Figure 3B). The anterior half of the hemicord and dural sac extended to the right inferior side and ended at the C6 level, toward the upper blind end of esophageal duplication. The posterior half joined the hemicord at the back and the C6 level (Figure 3C). The anterior half of the anterior hemicord was directed downwards, encompassed by the dural sac, with a thin CSF-filled track extending downwards. The end was directed to the upper blind end of the esophageal duplication observed in the UGI series (Figure 3D). The possibility of a form of neurenteric canal was suspected. At the cervicothoracic junction, there was a division of the medulla (diplomyelia) diverging laterally to the left and right, from which nerve roots of the upper limbs emerged. A bony septum or fibrous band between the two hemicords is not visible. No abnormal findings were observed during the nerve conduction study and needle electromyography. A gene study conducted as a work-up for multiple anomalies confirmed that the karyotype was normal (46,XY), but arr[GRCh37] 12p12.3(18,681,464-18,741,293)x1 was identified, which was classified as a variant of uncertain significance (VUS), according to the 2020 ACMG/ClinGen guidelines. The 60 kb loss detected in this section includes a part of the coding sequence of the PIK3C2G gene. However, the gene-phenotypic association of the PIK3C2G gene is unknown [11,12].

### 2.4. Management and Follow-Up

After multidisciplinary collaboration, careful follow-up and conservative treatment were planned. At 12 months, the patient had developmental delay (Table 1), and torticollis showed little improvement (Figure 4). No neurological abnormalities have been observed. The patient plans to undergo surgery for the cervical spine fusion anomaly. No clear genetic cause has been identified for multiple anomalies, and whole-genome sequencing for the patient and his parents is in progress.

## 3. Discussion

A defect in primary neurulation during embryological development of the spinal cord can lead to SCM. The incomplete understanding of embryologic development and scarcity of documented SCM cases has resulted in several nomenclatures (diastematomyelia, diplomyelia, and dimyelia) and hypotheses describing the form of spinal dysraphism since 1837. However, most were not widely accepted until Pang et al. [4] in 1992 introduced a unifying theory of embryogenesis and proposed a simple and widely recognized classification system. The theory suggests that SCM can be categorized into two types: Type I SCM (diastematomyelia) consists of two hemicords in two individual dural tubes separated by a bony spur. Type II (diplomyelia) corresponds to two hemicords surrounded by a single dural sac separated by a fibrous band. Several hypotheses have been proposed to describe the mechanism of SCM, including neural tube defects, but none have been conclusively established. Spinal axis formation, beginning in the first trimester of fetal development, starts with differentiation of the germ layers [13]. This process, referred to as primary neurulation, includes formation of three distinct germ cell layers (ectoblast, mesoblast, and endoblast) and the notochord. Derived from the endoderm, the notochord plays a central role in the development of the vertebral column, including axis constitution and signaling. During primary neurulation, the notochord induces neural plate formation at the ectoblast level, which subsequently invaginates to form the neural tube, the origin of the spinal cord. The vertebral column is also constructed during this period. Somites, embryonic paraxial mesoderms, form part of the vertebral column instructed by signals from the neural tube and the notochord. The cells in the dorsal walls of the somites eventually migrate around the neural tube to form the vertebral arches. Cervical SCM occurs during primary neurulation. According to Pang et al. [4], a lesion on the medial line from adherence between the endoderm and the ectoderm creates an accessory neurenteric canal (Figure 5A). This adhesion blocks the midline integration of mesodermal progenitor cells and results in a duplication of the notochord. As a result, neurulation occurs in two separate notochords, which leads to the subsequent induction of two hemicords (Figure 5B). He also suggested that the mesenchyme condenses around the endodermal-ectodermal adhesion to form an endomesenchymal tract (Figure 5C). Primitive mesenchymal and meningeal progenitor cells enter the endomesenchymal tract and subsequently mature into bone and dura (Figure 5D). When separated into two hemicords by a bony spur, it has its dural sheath (Type I SCM), and when divided into two hemicords by a fibrous septum, it is surrounded by the same dural sheath (Type II SCM). Although the factors determining type I and type II formation are still controversial, both types can cause spinal cord tethering. Our case is not perfectly represented by the types described by Pang because there was no osteocartilaginous septum or fibrous band longitudinally separating the spinal cord at the midline. This case showed the complete division of the spinal cord from dorsal to ventral. Furthermore, given the end of the isolated cord in this case we feel it is worth reporting. We hypothesized that the neurenteric cyst caused cord splitting during the embryonic period, moving toward the end of the isolated anterior division of the hemicord. This case (1) had a split cord malformation accompanied by cervical spine fusion anomaly, (2) consisted of a single dural sac, and (3) had no bony septum; thus, this case is considered to be closer to type II SCM than type I. Previous reports on SCM have differed from our case. David et al. [13] described a few cases of vertebral abnormalities associated with cervical SCM and proposed an embryological hypothesis of anterior or posterior arch defects. In the case of the posterior defect, it was suggested due to a defect in the closing process of the neural tube. On the other hand, in the anterior defect, It was suggested that endoderm invagination prevents the union of sclerotomes. In contrast, the anterior defect was thought to be a fusion problem of the sclerotomes by the abnormal invagination process of the endoderm. Andro et al. [14] reported a case of Klippel–Feil syndrome, in which the patient had SCM, associating diplomyelia, and cervicothoracic vertebral malformation. The child had two separate hemicords with an anterior and posterior bony defect, which was not seen in our case. Some genes are reportedly related to vertebral malformations. The PAX1 gene is associated with the vertebral fusion anomaly and Klippel–Feil syndrome [15,16], whereas Hox genes have been linked to variations in cervical vertebrae [17]. The long-term prognosis of SCM and the appropriate timing of surgery remain controversial. Proctor and Scott [18] reviewed 16 patients with SCM (11: SCM type 1, 5: SCM type 2) in whom surgery was performed. The mean age at surgery was 11 years, with an aim to remove the fibrous or bone septum and resect any local spinal cord attachments that were causing tethering. Although most patients tolerated surgery well, two patients with neurological deterioration showed rethethering. Furthermore, despite neurological improvement, including bladder and bowel function after surgery, vertebral column deformities gradually progressed, requiring subsequent spinal fusion. Another study [19] that analyzed patients with SCM who underwent surgery reported that 11 patients needed surgical correction for scoliosis. Given the complexity of our SCM case without bony spur or septum with accompanying vertebral fusion anomaly, meticulous and multidisciplinary discussion is required to determine the optimal timing of surgery.

## 4. Conclusions

A cervical spine X-ray should always be performed to rule out cervical vertebral segmentation anomaly, despite its rarity, when assessing a patient with torticollis. The conservative treatment of physical therapy is insufficient to manage torticollis when SCM with a vertebral anomaly is present. However, it can help to elongate connective tissue around the cervical spine. Since neurological deterioration and scoliosis can occur, close follow-up is needed in these patients. The optimal timing of surgery and management should be individualized and discussed by a multidisciplinary team.

## Figures and Tables

**Figure 1 children-09-01085-f001:**
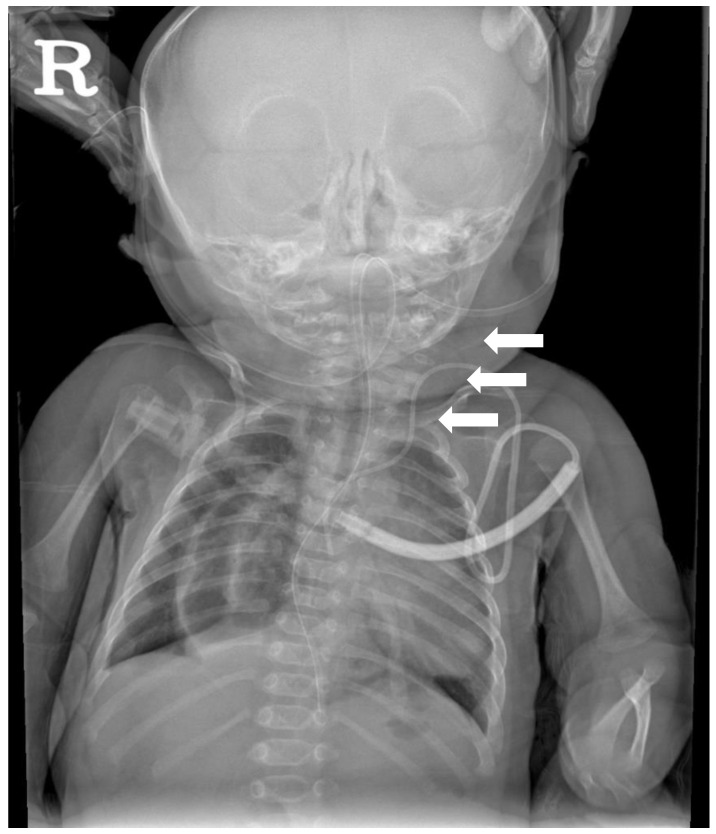
Cervical spine X-ray AP view performed at 2 months of age showing the extensive segmentation-fusion anomaly involving the cervical spine (white arrows).

**Figure 2 children-09-01085-f002:**
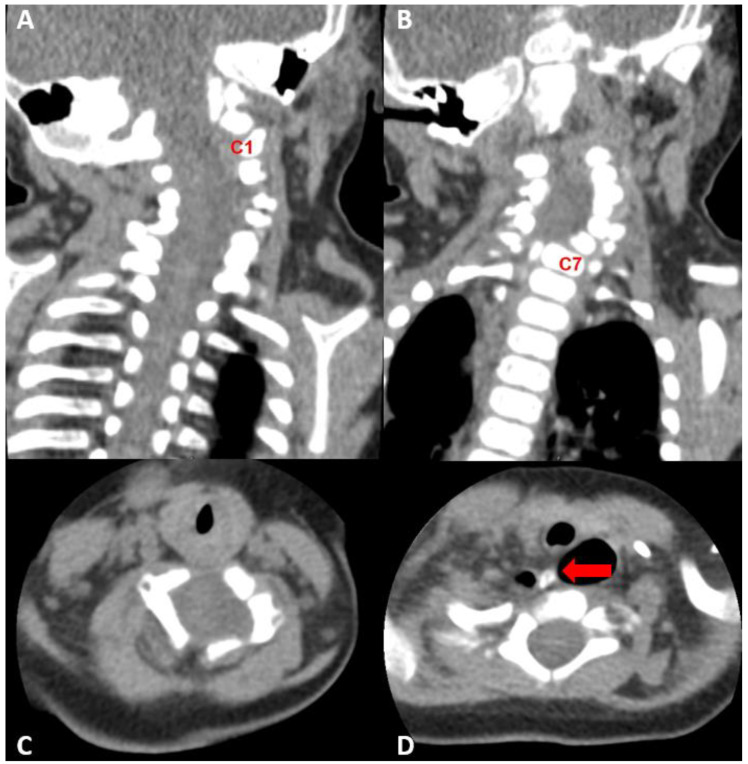
Cervical spine CT performed at 4 months of age. (**A**,**B**) Extensive segmentation-fusion anomaly with anterior and posterior bony defect in the C1–6 vertebrae (C1 and C7 vertebra were marked for orientation), (**C**) C4-level axial view, widened central canal was observed, (**D**) bony spurs arising from the C6 vertebra, directing to the mediastinum and abutting upper blind end of the esophageal duplication cyst.

**Figure 3 children-09-01085-f003:**
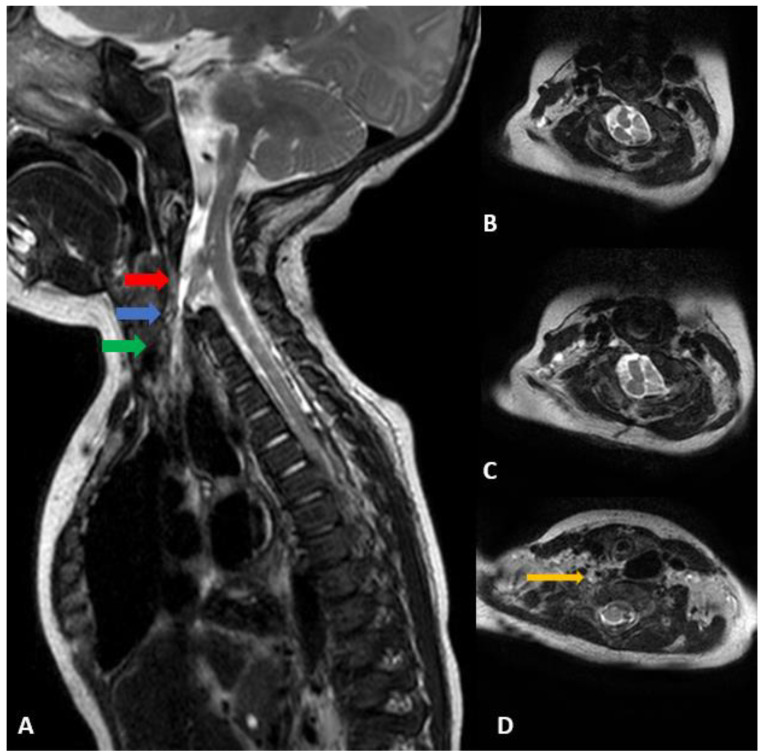
(**A**) Cervical spine sagittal T2 MRI performed at 4 months of age. Extensive segmentation-fusion anomaly with an anterior bony defect in the C vertebrae, anterior half of the hemicord and dural sac extending to the right inferior side, toward the upper blind end of esophageal duplication. (**B**) From the C3 level, the spinal cord splits forward and backward, showing two hemicords. (**C**) The anterior half of the hemicord branches slightly to the right side and ends at the C6 level, and the posterior half joins the hemicord at the back and C6 level. (**D**) The anterior half of the anterior hemicord goes down and the dural sac bulges around it. A thin CSF-filled track extends downwards, and the end is directed to the upper blind end of the esophageal duplication observed in the UGI series (orange arrow). (**B**) Axial view of the red arrow, (**C**) axial view of the blue arrow, (**D**) Axial view of the green arrow.

**Figure 4 children-09-01085-f004:**
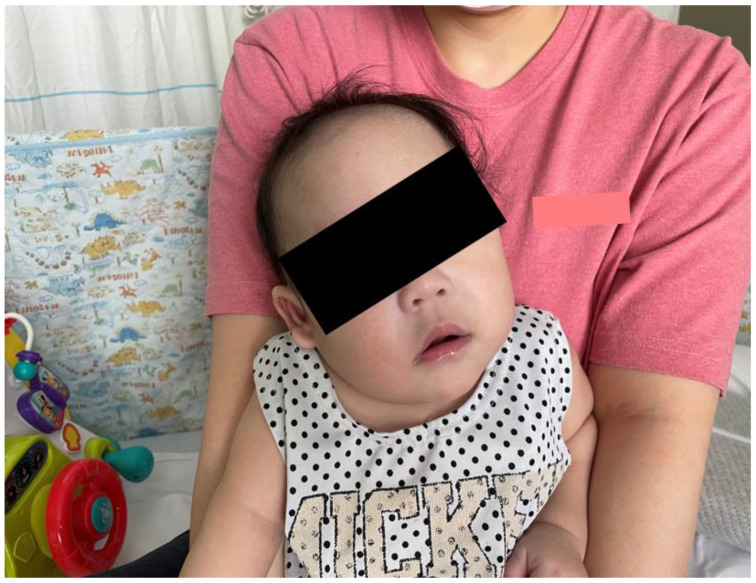
Symptom of torticollis left at 12 months old.

**Figure 5 children-09-01085-f005:**
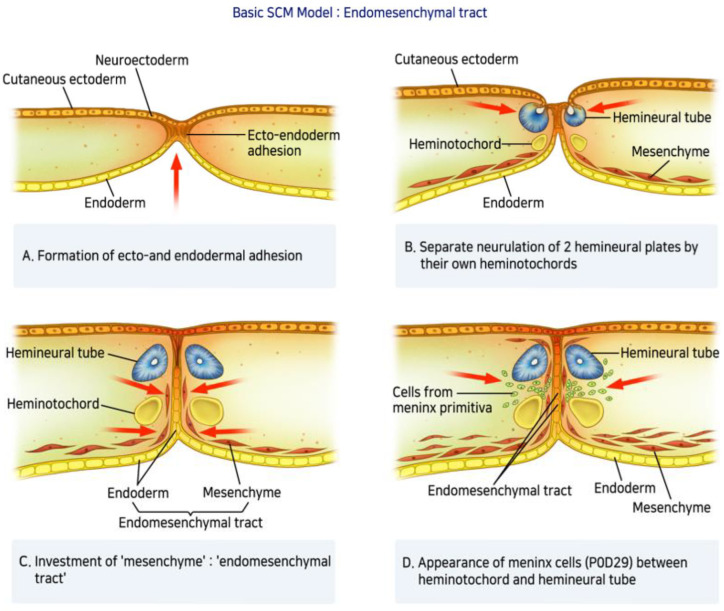
Basic SCM Model: Endomesenchymal tract—(**A**) A lesion on the medial line from adherence between the endoderm and the ectoderm creates an accessory neurenteric canal. (**B**) This adhesion blocks the midline integration of mesodermal progenitor cells and results in a duplication of the notochord. As a result, neurulation occurs in two separate notochords, which leads to the subsequent induction of two hemicords. (**C**) The mesenchyme condenses around the endodermal-ectodermal adhesion to form an endomesenchymal tract. (**D**) Primitive mesenchymal and meningeal progenitor cells enter the endomesenchymal tract and subsequently mature into bone and dura.

**Table 1 children-09-01085-t001:** Results of Denver Developmental Screening Test.

Results at	4 Months	7 Months	12 Months
Personal-Social (months)	3	5	11
Fine Motor-Adaptive (months)	3	7	11
Language (months)	3	6	10
Gross Motor (months)	1	5	7

## Data Availability

Not applicable.

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
