# Peer review of "An Unusual Case of Torticollis: Split Cord Malformation with Vertebral Fusion Anomaly: A Case Report and a Review of the Literature"

_children, 2022, doi:10.3390/children9071085_

Round 1
Reviewer 1 Report
An excellent, documented case report of the initial existence of torticollis, with clear guidelines for radiological diagnostics that can detect other anomalies of the spinal cord and cervical vertebral malformation.You are given recommendations and possibilities for treatment, which is multidisciplinary
The case report, which is very significant, shows the initial existence of torticollis, and in the further diagnostic algorithm it shows spinal cord malformation, associating neurenteric cyst, and cervical vertebral malformation.
A clinical entity shown from an embryological and genetic aspect, a detailed clinical picture, documented by a radiological finding with a discussion that provides guidelines for treatment that is multidisciplinary.
It indicates the importance and necessity of radiological diagnostics in children with the existence of torticollis, and the insufficiency of rehabilitation treatment when SCM with a vertebral anomaly is present.
Author Response
Thank you so much for thoroughly reading our article and giving your kind comments despite your busy schedule.
I am so impressed with your thoughtful comment. Thank you once again.
Best regards,
Eun Jae Ko MD, PhD.
Reviewer 2 Report
This article describes a case of torticollis with split cord malformation with vertebral fusion anomaly. The strengths of this manuscript are that it is well written. It is easy to follow the results and the workflow the authors followed to achieve the conclusions.
1. In Figure 2, Is there a 3D-reconstructed CT image in this case? 3D images are easier to follow visually.
2. arr[GRCh37] 12p12.3(18,681,464-18,741,293)x1. What genes are present in this region? Are any of them potentially responsible genes?
3. Are there any complications of morphological abnormalities of the brain?
Author Response
Thank you so much for thoroughly reading our article and giving your kind comments despite your busy schedule.
Point 1: In Figure 2, Is there a 3D-reconstructed CT image in this case? 3D images are easier to follow visually.
Response 1:
Unfortunately, there was no 3D image in C-spine CT. In our hospital, a protocol for performing 3D images on the whole spine images is in progress. As you said, having a 3D image will be much more helpful for tracking and understanding. If we see this patient or a similar case in the future, we will perform 3D CT together. Thank you so much for your thoughtful comments.
Point 2: arr[GRCh37] 12p12.3(18,681,464-18,741,293)x1. What genes are present in this region? Are any of them potentially responsible genes?
Response 2:
A 60kb loss was detected in a part of the coding sequence of the PIK3C2G gene. However, the gene-phenotypic association of the PIK3C2G gene is unknown. The gene study we conducted and the data related to the detected PIK3C2G gene are described below.
â—ˆ Detailed variant information and Interpretation
1) arr[GRCh37] 12p12.3(18,681,464-18,741,293)x1, 60kb loss, Uncertain significance (VUS)
- 1 Genes
- 1 OMIM genes: PIK3C2G
- The detected 60kb loss includes a part of the coding sequence of the PIK3C2G gene.
- In this section, there is no haploinsufficiency/triplosensitivity gene organized in the ClinGen Dosage Sensitivity Map, and it is not an established benign region. In the disease group DB and population DB, the CNV overlapping with the patient's CNV was reported as Benign and VUS.
- There is no known gene-phenotypic association of the PIK3C2G (*609001) gene.
2) No clinically significant Absence of heterozygosity (AOH) greater than 5 Mb was detected.
â—ˆ Reference
- Cloning and characterization of a novel class II phosphoinositide 3-kinase containing C2 domain. Biochem. Biophys. Res. Commun. 244: 531-539, 1998.
- Technical standards for the interpretation and reporting of constitutional copy-number variants: a joint consensus recommendation of the American College of Medical Genetics and Genomics (ACMG) and the Clinical Genome Resource (ClinGen). Genet Med. 2020;22(2):245-257
Point 3: Are there any complications of morphological abnormalities of the brain?
Thanks for the excellent point. Brain MRI was not performed as there is currently no neurologic deficit and is following almost normal development. However, there is an anomaly in upper-level C-spines, so brain MRI is considered necessary in the future.
(In Bayley's scales of infant development performed at 14 months of age, the mental development index was 75, and the psychomotor development index was 56.)
